# SPECTRAL SELF-SUPERVISED FEATURE SELECTION

## ABSTRACT

Selecting a meaningful subset of features from high-dimensional observations in unsupervised settings can significantly improve the accuracy of downstream analysis tasks such as clustering or dimensionality reduction and provide insight into the sources of heterogeneity in a given dataset. In this paper, we derive a self-supervised graph-based approach for unsupervised feature selection. The core of our method is the robust computation of pseudo-labels by applying simple processing steps to the graph Laplacian's eigenvectors. The subset of eigenvectors used for computing pseudo-labels is chosen according to a model stability criterion. The importance of each feature is then measured by training a surrogate model to predict the pseudo-labels from the observations. We show that our method is robust to challenging scenarios, such as the existence of outliers and complex substructures. Our approach's efficacy is demonstrated through experiments on real-world datasets, showing its robustness across multiple domains and particular effectiveness on biological datasets.

## 1 INTRODUCTION

Sampling technology improvements enable scientists across many disciplines to acquire numerous variables from biological or physical systems. One of the key challenges in real-world scientific data is noisy, information-poor, or nuisance features. While such features could be mildly harmful to supervised learning, they could dramatically affect the downstream analysis (e.g., clustering or manifold learning) in the unsupervised setting (Mahdavi et al., 2019). To perform reliable data-driven scientific discovery, there is a growing need for unsupervised feature selection schemes that may enhance the latent signal of interest by removing nuisance variables.

Unsupervised feature selection (UFS) methods aim to identify a subset of informative features and thus improve the outcome of downstream analysis tasks such as clustering and manifold learning. For unsupervised tasks, and in the lack of labels such as a specific cluster or the value of a latent parameter, the downstream task cannot be used to drive the selection of features. Therefore, most UFS methods use a label-free criterion that is assumed to correlate with the downstream task. For example, many UFS schemes rely on a reconstruction prior (Li et al., 2017) and seek a subset of features that can be used to reconstruct the entire set of features. Autoencoders (AE) are used to learn a reduced representation of the data, and a sparsification penalty is introduced to force the AE to remove redundant features. This idea was implemented with several types of sparsity-inducing regularizers, including $\ell_{2,1}$ based (Chandra & Sharma, 2015; Han et al., 2018), relaxed $\ell_0$ (Balın et al., 2019; Shaham et al., 2022) and more.

Another widely used criterion for UFS is feature smoothness. Here, the hypothesis is that the structure of interest (clusters or a manifold) is typically a low-dimensional or low-rank and can be captured using the graph Laplacian matrix (Ng et al., 2001). Then, the smoothness of features is measured based on the Rayleigh quotient of the Laplacian, a measure known as the Laplacian Score (LS) (He et al., 2005). A feature that is smooth with respect to the graph is assumed to be associated with the main underlying data structures. Many other UFS methods use a graph to select informative features; these include Nonnegative Discriminative Feature Selection (NDFS) (Li et al., 2012), which performs feature selection and spectral clustering simultaneously, and its extension Li & Tang (2015) which adds a loss term that punishes the joint selection of correlated features. Other graph-based approaches include Li et al. (2018); Roffo et al. (2017); Zhu et al. (2017; 2020); Xie et al. (2023).

Embedded unsupervised feature selection schemes aim to cluster the data while simultaneously removing irrelevant features. Examples include Wang et al. (2015), which performs the selection directly on the clustering matrix, and Zhu & Yang (2018), which learns feature weights while maximizing the distance between clusters. In recent years, several works have derived self-supervised learning methods for feature selection. The key idea is to design a supervised type learning task with pseudo-labels that do not require human annotation. A seminal work that is based on this paradigm is Multi-Cluster Feature Selection (MCFS) (Cai et al., 2010). MCFS uses the eigenvectors of the graph Laplacian as pseudo-labels and learns the informative features by optimizing over an $\ell_1$ regularized least squares problem. More recently, Lee et al. (2021) used self-supervision with correlated random gates to enhance the performance of feature selection.

In this work, we present a spectral self-supervised scheme for feature selection. The main idea is to leverage the eigenvectors of the graph Laplacian selectively and discriminatively. This process is implemented through a multistage approach. We first generate robust discrete pseudo-labels from the eigenvectors, followed by filtering them based on a stability measure. We then fit flexible surrogate classification models on the selected eigenvectors and query the models for feature scores. Using these components, we can identify informative features that are shown to be effective for clustering on real-world datasets.

## 2 PRELIMINARIES

### 2.1 LAPLACIAN SCORE AND REPRESENTATION-BASED FEATURE SELECTION

Computing a graph-based representation for a set of high-dimensional observations has become standard practice for tasks in unsupervised learning. In manifold learning, methods such as ISOM-PAS (Tenenbaum et al., 2000), LLE (Roweis & Saul, 2000), Laplacian eigenmaps (Belkin & Niyogi, 2003), and diffusion maps (Coifman & Lafon, 2006) compute a low-dimensional representation that is associated with the manifold's latent structure. In spectral clustering, a set of points is partitioned by applying the $k$-means algorithm to the leading Laplacian eigenvectors (Ng et al., 2001).

In graph methods, each node $v_i$ corresponds to one of the observations $\boldsymbol{x}_i \in \mathbb{R}^p$. The weight $W_{ij}$ between two nodes $v_i, v_j$ is computed based on some kernel function $K(\boldsymbol{x}_i, \boldsymbol{x}_j)$. For example, the popular Gaussian kernel is equal to,

$$K(\boldsymbol{x}_i, \boldsymbol{x}_j) = \exp\left( -\frac{\|\boldsymbol{x}_i - \boldsymbol{x}_j\|^2}{2\sigma^2} \right).$$

Where the parameter $\sigma$ determines the bandwidth of the kernel function. Let $\boldsymbol{D}$ be a diagonal matrix with the degree of each node in the diagonal, such that $D_{ii} = \sum_j W_{ij}$. The unnormalized graph Laplacian matrix is equal to

$$\boldsymbol{L} = \boldsymbol{D} - \boldsymbol{W}.$$

For any vector $\boldsymbol{v} \in \mathbb{R}^n$ we have the following equality (Von Luxburg, 2007),

$$\boldsymbol{v}^T \boldsymbol{L} \boldsymbol{v} = \frac{1}{2} \sum_{i,j} \left( v_i - v_j \right)^2 W_{i,j}. \tag{1}$$

The quadratic form in equation 1 gives rise to a notion of graph *smoothness*. (Ricaud et al., 2019; Shuman et al., 2013). A vector is smooth with respect to a graph if it has similar values on pairs of nodes connected with an edge with a significant weight. This notion underlies the Laplacian score suggested as a measure for unsupervised feature selection (He et al., 2005). Let $\boldsymbol{f}_m \in \mathbb{R}^n$ denote the values of the $m$-th feature for all observations. The Laplacian score $s_m$ is equal to,

$$s_m = \boldsymbol{f}_m^T \boldsymbol{L} \boldsymbol{f}_m = \frac{1}{2} \sum_{i,j} \left( f_{m,i} - f_{m,j} \right)^2 W_{ij}. \tag{2}$$

A low score indicates that a feature is smooth with respect to the computed graph and thus strongly associated with the latent structure of the high-dimensional data $\boldsymbol{x}_1, \ldots, \boldsymbol{x}_n$. The notion of the Laplacian score has been the basis of several other feature selection methods as well (Lindenbaum et al., 2021; Shaham et al., 2022; Zhu et al., 2012).

Let $\boldsymbol{v}_i, \lambda_i$ denote the $i$-th smallest eigenvector and eigenvalue of the Laplacian $\boldsymbol{L}$. A slightly different interpretation of equation 2 is that the score for each feature is equal to a weighted sum of its

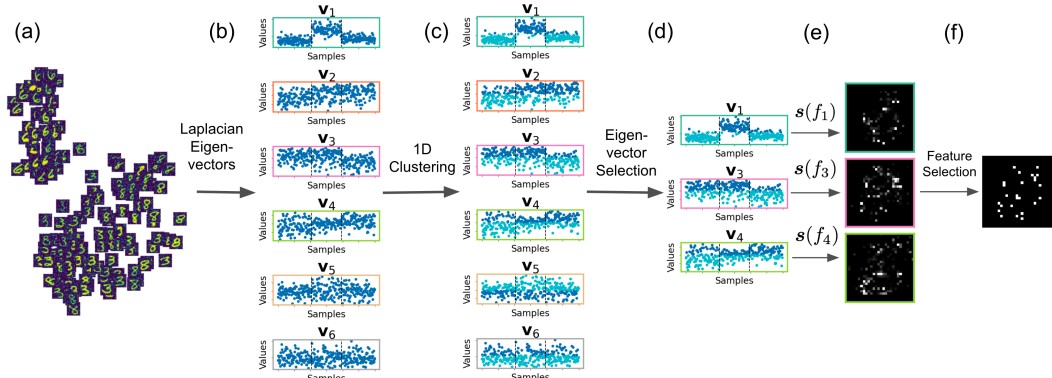

Figure 1: Outline of the Spectral Self-supervised Unsupervised Feature Selection algorithm: (a) A tSNE scatter plot of noisy MNIST digits (3, 6, 8). (b) The six leading eigenvectors of the graph Laplacian. Samples are ordered according to the identity of the digit. (c) The six leading eigenvectors colored by the output of the $k$-medoids algorithm, which defines the pseudo-labels $\boldsymbol{y}_i^*$. (d) Selecting three eigenvectors with the most stable model $h_i(X; y)$. (e) Computing feature scores for each model $f_i(X; y)$. (f) Aggregation of all feature scores across eigenvectors.

correlation with the eigenvectors, such that

$$s_m = \sum_{i=1}^n \lambda_i (\boldsymbol{f}_m^T \boldsymbol{v}_i)^2.$$

A potential drawback of the Laplacian score is its dependence on a large number of eigenvectors. This may reduce its stability in measuring a feature's importance to the data's main structures. To overcome this limitation, Zhao & Liu (2007) derived an alternative score based only on a feature's correlation to the leading Laplacian eigenvectors. A related, more sophisticated approach is Multi-Cluster Feature Selection (MCFS) (Cai et al., 2010), which computes the solutions to the generalized eigenvector problem $\boldsymbol{Lv} = \lambda \boldsymbol{Dv}$. The leading eigenvectors are then used as pseudo-labels for a regression task with $l_1$ regularization. Specifically, MCFS applies Least Angle Regression (LARS) (Efron et al., 2004) to obtain, for each leading eigenvector $\boldsymbol{v}_i$, a sparse vector of coefficients $\boldsymbol{\beta}^i \in \mathbb{R}^p$. A feature score is computed by maximizing the absolute values of its corresponding coefficient,

$$s_j = \max_i |\beta_j^i|.$$

The output of MCFS is the set of features with the highest score. In the next section, we derive Spectral Self-supervised Feature Selection (SSFS), which improves upon the MCFS algorithm in several critical aspects.

## 3 SPECTRAL SELF-SUPERVISED FEATURE SELECTION

### 3.1 RATIONALE

As its title suggests, MCFS aims to uncover features that separate clusters in the data. Let us consider an ideal case where the observations are partitioned into $k$ well-separated clusters, denoted $A_1, \ldots, A_k$, such that the weight matrix $W_{ij} = 0$ if $\boldsymbol{x}_i, \boldsymbol{x}_j$ are in separate clusters. Let $\boldsymbol{e}^i$ denote an indicator vector for cluster $i$ such that

$$e_j^i = \begin{cases} 1/\sqrt{|A_i|} & j \in A_i \\ 0 & \text{o.w,} \end{cases}$$

where $|A_i|$ denotes the size of cluster $A_i$. In this scenario, the zero eigenvalue of the graph Laplacian has multiplicity $k$, and the corresponding eigenvectors are equal, up to a rotation matrix, to a matrix $E \in \mathbb{R}^{n \times d}$ whose columns are equal to $\boldsymbol{e}^1, \ldots, \boldsymbol{e}^k$. In such a case, the $k$ leading eigenvectors are indeed suitable for use as pseudo-labels for the feature selection task. Assuming that the clusters are amenable to a linear separation, the MCFS algorithm should provide highly informative features in terms of cluster separation.

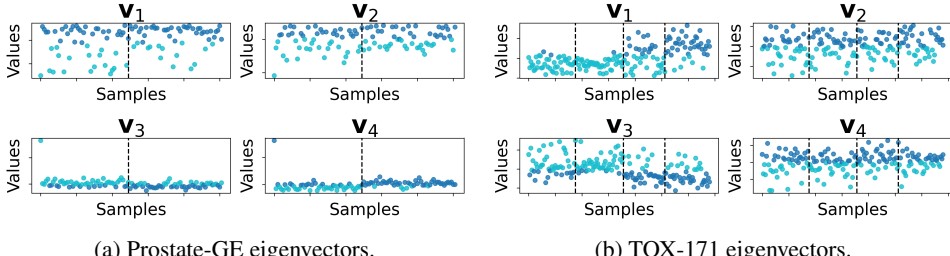

(a) Prostate-GE eigenvectors.  (b) TOX-171 eigenvectors.

Figure 2: The first four Laplacian eigenvectors of two real datasets. Samples are sorted according to the real class label and colored by the outcome of a one-dimensional $k$-medoids per eigenvectors. The vertical bar indicates the separation between the classes. In Prostate-GE, $v_4$ is the most informative to the class labels, and an outlier can be seen on the upper left in the third and fourth eigenvectors. In TOX-171, $v_3$ is more informative to the class labels than $v_2$.

However, the cluster separation can be far from perfect in many applications. In such cases, using the leading $k$ eigenvectors as pseudo-labels for regression may be suboptimal. Let us describe a couple of common scenarios:

- Complex high-dimensional datasets may contain various substructures that are not the primary interest but manifest in the top eigenvectors. In contrast, the main structure of interest appears only later in the spectrum. For illustration, consider the MNIST dataset visualized via tSNE in Figure 1(a). The data contains images of $3, 6$ and $8$. Panel (b) shows the elements of the six leading eigenvectors of the graph Laplacian matrix, sorted by their corresponding digits. The leading eigenvector shows a clear gap between images of digit $6$ and the rest of the data. However, there is no clear separation between digits $3$ and $8$. Indeed, the next eigenvector is not associated with such a separation. Applying feature selection with this eigenvector may produce spurious features irrelevant to separating the two digits. This scenario is prevalent in the real datasets used in the experimental section. For example, Figure 2a shows four eigenvectors of a graph computed from observations containing the genetic expression data from prostate cancer patients and controls (Singh et al., 2002). The leading two eigenvectors, however, are not associated with the patient-control separation.

- The leading eigenvectors may be affected by outliers. For example, an eigenvector may indicate a small group of outliers separated from the rest of the data. This phenomenon can also be seen in the third and fourth vectors of the Prostate-GE example in Figure 2a. While the fourth eigenvector separates the categories, it is corrupted by outliers and, hence, unsuitable for use as pseudo-labels in a classical regression task, as it might highlight features associated with the outliers.

- The relation between important features and the separation of clusters may be highly non-linear. In such cases, applying linear regression models to obtain feature scores may be too restrictive.

Motivated by the above scenarios, we derive Spectral Self-supervised Feature Selection (SSFS). We explain our approach in detail in the following two sections.

## 3.2 EIGENVECTOR PROCESSING AND SELECTION

**Generating binary labels.** Given the Laplacian eigenvectors $V = (v_1, ..., v_d)$, our goal is to generate pseudo-labels that are highly informative to the cluster separation in the data. To that end, for each eigenvector $v_i$, we compute a binary label vector $y_i^*$ (pseudo-labels) by applying a one-dimensional $k$-medoids algorithm (Kaufman & Rousseeuw, 1990) to the elements of $v_i$. In contrast to $k$-means, in $k$-medoids, the cluster centers are set to one of the input points, which makes the algorithm robust to outliers. In Figure 2, the eigenvectors are colored according to the output of the $k$-medoids. After binarization, the fourth eigenvector of the Prostate-GE dataset is highly indicative of the category. The feature selection is thus based on a classification rather than a regression task, which is more aligned with selecting features for clustering. In Section 4.2 we show the impact of the binarization step on multiple real-world datasets.

**Eigenvector selection.** Selecting $k$ eigenvectors according to their eigenvalues may be unstable in cases where the eigenvalues exhibit a small spectral gap. For SSFS, we derive a robust criterion for selecting informative eigenvectors that is based on the stability of a model learned for each vector. Formally, we consider a surrogate model $h : \mathbb{R}^p \to \mathbb{R}$, and a feature score function $\boldsymbol{s}(h) \in \mathbb{R}^p$, where $p$ denotes the number of features. For example, $h$ can be the logistic regression model $h(\boldsymbol{x}) = \sigma(\boldsymbol{\beta}^T \boldsymbol{x})$. In that case, a natural score function is the absolute value of the coefficient vector $\boldsymbol{\beta}$. For each eigenvector $\boldsymbol{v}_i$, we train a model $h_i$ on $B$ (non-mutually exclusive) subsets of the input data $\boldsymbol{X}$ and the pseudo-labels $\boldsymbol{y}_i^*$. We then estimate the variance of the feature score function, for every feature $m \in \{1, ..., p\}$:

$$\widehat{\mathrm{Var}}(s_m(h_i)) = \frac{1}{B-1} \sum_{b=1}^{B} (s_m(h_{i,b}) - \bar{s}_m(h_i))^2.$$

This procedure is similar (though not identical) to the d-delete Jackknife method for variance estimation (Shao & Wu, 1989). We keep, as pseudo-labels, the $k$ binarized eigenvectors with the lowest sum of variance, $\hat{\mathcal{S}}_i = \sum_{m=1}^{p} \widehat{\mathrm{Var}}(s_m(h_i))$. We denote the set of selected eigenvectors by $I$. A pseudo-code for the pseudo-labels generation and eigenvector selection steps appears in Alg. 1.

### 3.3 Feature selection

For the feature selection step, we train $k$ models, denoted $\{f_i \mid i \in I\}$, to predict the selected binary pseudo-labels based on the original data. Similarly to the eigenvector selection step, each model is associated with a feature score function $\boldsymbol{s}(f_i)$. The features are then scored according to the following maximum criterion,

$$\mathrm{score}(m) = \max_{i \in I} s_m(f_i).$$

Finally, the features are ranked by their scores, and the top-ranked features are selected for the subsequent analysis. The choice of model for this step can differ from that used in the eigenvector selection step, allowing for flexibility in the modeling approach (see Section 3.4 for details). Pseudo-code for SSFS appears in Algorithm 2.

### 3.4 Choice of Surrogate Models

Our algorithm is compatible with any supervised model capable of providing feature importance scores. We combine the structural information from the graph Laplacian with the capabilities of various supervised models for unsupervised feature selection. Empirical evidence supports the use of more complex models such as Gradient-Boosted Decision Trees for various complex, real-world datasets (McElfresh et al., 2023; Chen & Guestrin, 2016). These models are capable of capturing complex nonlinear relationships, which we leverage by training them on pseudo-labels derived from the Laplacian's eigenvectors. For example, for eigenvector selection, one can use a simple logistic regression model for fast training on the resampling procedure and a more complex gradient boosting model such as XGBoost (Chen & Guestrin, 2016) for the feature selection step.

## 4 Experiments

### 4.1 Evaluation on Real World Datasets

**Data and experiment description:** We applied SSFS to eight real-world datasets from various domains. Table 1 gives the number of features, samples, and the number of different classes among the observations for each dataset. All datasets are available online [1].

We compare the performance of our approach to the following alternatives: (i) standard Laplacian score (LS) (He et al., 2005), (ii) Multi-Cluster Feature Selection (MCFS) (Cai et al., 2010), (iii) Nonnegative Discriminative Feature Selection (NDFS), (Li et al., 2012), (iv) Unsupervised Discriminative Feature Selection (UDFS) (Yang et al., 2011), and (v) Laplacian Score-regularized Concrete Autoencoder (LS-CAE) (Shaham et al., 2022). To measure the accuracy of each method, we use the following criterion, previously used in several UFS papers (Li et al., 2012; Cai et al., 2010). We select the top 2, 5, 10, 20, 30, 40, 50, 100, 150, 200, 250, and 300 features for each method. Then,

---

[1] https://jundongl.github.io/scikit-feature/datasets.html

---

**Algorithm 1** Pseudo-code for Eigenvector Selection and Pseudo-labels Generation

---

**Require:** Dataset $\boldsymbol{X} \in \mathbb{R}^{n \times p}$ (with $n$ samples and $p$ features), number of eigenvectors to select $k$, number of eigenvectors to compute $d$, surrogate models $H = \{h_i \mid i \in [d]\}$, feature scoring function $\boldsymbol{s} : \mathcal{F} \to \mathbb{R}^p$, number of resamples $B$

1: Initialize an empty list for the pseudo pseudo-labels $\mathcal{Y}^*$ and an empty list for the sums of features variance $\hat{\mathcal{S}}$
2: Compute the significant $d$ eigenvectors of the Laplacian of $\boldsymbol{X}$: $\boldsymbol{V} = (\boldsymbol{v}_1, ..., \boldsymbol{v}_d)$
3: **for** $i = 1$ to $d$ **do**
4:     Binarize the eigenvector $\boldsymbol{v}_i$ using $k$-medoids to obtain $\boldsymbol{y}_i^*$, and append to $\mathcal{Y}^*$
5:     **for** $b = 1$ to $B$ **do**
6:         Subsample $((\boldsymbol{X})_b, (\boldsymbol{y}_i^*)_b)$ from $(\boldsymbol{X}, \boldsymbol{y}_i^*)$
7:         Fit the model $h_{i,b}$ to $((\boldsymbol{X})_b, (\boldsymbol{y}_i^*)_b)$
8:     **end for**
9:     **for** $m = 1$ to $p$ **do**
10:         Estimate the variance of the $m$-th feature score:

$$\widehat{\mathrm{Var}}(s_m(h_i)) = \frac{1}{B-1} \sum_{b=1}^{B} (s_m(h_{i,b}) - \bar{s}_m(h_i))^2$$

11:     **end for**
12:     $\hat{\mathcal{S}}_i = \sum_{m=1}^{p} \widehat{\mathrm{Var}}(s_m(h_i))$
13:     $\hat{\mathcal{S}} \leftarrow \hat{\mathcal{S}} \cup \{\hat{\mathcal{S}}_i\}$
14: **end for**
15: Select the indices of the $k$ smallest elements in $\hat{\mathcal{S}}$ and store in $I$
16: **return** $\mathcal{Y}^*, I$

---

**Algorithm 2** Pseudo-code for Spectral Self-supervised Feature Selection (SSFS)

---

**Require:** Dataset $\boldsymbol{X} \in \mathbb{R}^{n \times p}$ (with $n$ samples and $p$ features) number of eigenvectors to select $k$, number of eigenvectors to compute $d$, surrogate eigenvector selection models $H = \{h_i \mid i \in [d]\}$, surrogate feature selection models $F = \{f_i \mid i \in [d]\}$, feature scoring function $\boldsymbol{s} : \mathcal{F} \to \mathbb{R}^p$, number of resamples $B$, number of features to select $\ell$.

1: Apply Algorithm 1 to obtain the pseudo-labels and the selected eigenvectors:
   $\mathcal{Y}^*, I = $ **EigenvectorPreprocessingAndSelection**$(\boldsymbol{X}, k, d, H, \boldsymbol{s}, B)$
2: **for** $i$ in $I$ **do**
3:     Fit the model $f_i$ on $(\boldsymbol{X}, \boldsymbol{y}_i^*)$
4:     Calculate the feature scores $\boldsymbol{s}(f_i)$
5: **end for**
6: **for** $m = 1$ to $p$ **do**
7:     Compute the final score for the $m$-th feature:

$$\text{score}(m) = \max_{i \in I} s_m(f_i)$$

8: **end for**
9: **return** a list of $\ell$ features with the highest score.

---

we apply $k$-means 20 times on the selected features and compute the average clustering accuracy, computed by (Cai et al., 2011):

$$\mathrm{ACC} = \max_{\pi} \frac{1}{N} \sum_{i=1}^{N} \delta(\pi(c_i), l_i),$$

where $c_i$ and $l_i$ are the assigned cluster and true label of the $i$-th data point, respectively, $\delta(x, y)$ is the delta function which equals one if $x = y$ and zero otherwise, and $\pi$ represents a permutation of the cluster labels. The optimization over $\pi$ can be carried out using the Kuhn-Munkres algorithm (Munkres, 1957). Table 2 shows, for each method, the highest average accuracy and the number of features for which it was achieved. Figure 3 shows for each dataset and method the clustering accuracy for the full range of selected features.

For SSFS, we use the following surrogate models:

- The eigenvector selection model $h_i$ is set to Logistic Regression with L2 regularization. We use scikit-learn's (Pedregosa et al., 2011) implementation with a default regularization value of $C = 1.0$. Feature scores are equal to the absolute value of the model's coefficients.

- The feature selection model $f_i$ is set to XGBoost classifier with *Gain* feature importance. We use the popular implementation by DMLC (Chen & Guestrin, 2016).

Table 1: Real-world datasets description.

| Dataset | Samples | Dim | Classes | Domain |
|---|---|---|---|---|
| COIL20 | 1440 | 1024 | 20 | Image |
| ORL | 400 | 1024 | 40 | Image |
| Yale | 165 | 1024 | 15 | Bio |
| ALLAML | 72 | 7129 | 2 | Bio |
| Prostate-GE | 102 | 5966 | 2 | Bio |
| TOX 171 | 171 | 5748 | 4 | Bio |
| Isolet | 1560 | 617 | 26 | Speech |
| GISETTE | 7000 | 5000 | 2 | Image |

Note that we employ the default hyper-parameters for all surrogate models as provided in their widely used implementations. However, it's worth noting that one can undoubtedly leverage domain knowledge to select surrogate models and hyperparameters better suited to the specific domain. In addition, for each dataset, SSFS selects $k$ from $d = 2k$ eigenvectors, where $k$ is the number of distinct classes in the data.

**Results.** SSFS ranks best in four of the eight datasets. The advantage over competing methods is particularly significant in the Yale, TOX-171, and Prostate-GE datasets. As discussed in Section 3.1, the Prostate-GE dataset contains several outliers. In addition, the fourth eigenvector is highly informative in terms of the class labels compared to the earlier eigenvectors. The ability of SSFS to deal with such challenging scenarios might explain its performance. For the other four datasets, while our method is not ranked first, its outcome is on par with the result of the leading method.

Table 2: Average clustering accuracy on benchmark datasets. The number of selected features yielding the best clustering performance is shown in parentheses, with the best method for each dataset highlighted in bold.

| Dataset | LS | MCFS | NDFS | UDFS | LS-CAE | SSFS |
|---|---|---|---|---|---|---|
| COIL20 | 61.9 (300) | **67.4 (300)** | 63.4 (200) | 61.9 (300) | 65.1 (100) | 67.1 (300) |
| GISETTE | 70.0 (250) | **70.7 (5)** | 58.3 (100) | 69.1 (50) | 66.4 (10) | 69.7 (150) |
| Yale | 43.9 (300) | 44.4 (300) | 43.5 (250) | 43.8 (50) | 45.1 (50) | **50.3 (100)** |
| TOX-171 | 51.3 (5) | 44.5 (5) | 47.3 (150) | 40.2 (250) | 46.1 (250) | **59.4 (100)** |
| ALLAML | 72.2 (200) | 75.0 (150) | **76.6 (2)** | 66.4 (50) | 63.6 (5) | 75.4 (100) |
| Prostate-GE | 58.8 (2) | 61.8 (100) | 58.8 (2) | 63.6 (50) | 63.7 (200) | **75.9 (10)** |
| ORL | 51.6 (300) | 57.0 (300) | 59.1 (300) | 57.3 (300) | 60.7 (300) | **61.1 (200)** |
| ISOLET | 48.9 (300) | 50.7 (300) | **63.1 (200)** | 44.6 (300) | 62.8 (300) | 59.9 (100) |
| Mean rank | 4.25 | 3.25 | 3.69 | 4.81 | 3.25 | **1.75** |
| Median rank | 4.5 | 3.5 | 3.5 | 5.0 | 2.5 | **1.5** |

## 4.2 ABLATION STUDY

In this section, we demonstrate the importance of the following components of SSFS: (i) eigenvector selection, (ii) self-supervision with nonlinear models as surrogate models, and (iii) the binarization of the Laplacian eigenvectors along with classifiers instead of regressors as surrogate models.

The ablation study is performed both on a synthetic dataset described in Section 4.2.1, and on the eight real datasets used for evaluation in Section 4.1.

### 4.2.1 SYNTHETIC DATA

We generate a synthetic dataset as follows: the first five features are generated from two isotropic Gaussian blobs; these blobs define the clusters of interest. Additional 45 nuisance features are generated according to a multivariate Gaussian distribution, with zero mean and a block-structured covariance matrix $\Sigma$, such that each block contains 15 features. The covariance elements $\Sigma_{i,j}$ are equal to $0.5$ if $i, j$ are in the same block, and to $0.01$ otherwise. We generated a total of 500 samples, see Appendix A.1.1 for further details. Figurse 4a and 4b show, respectively, a scatter plot of the first

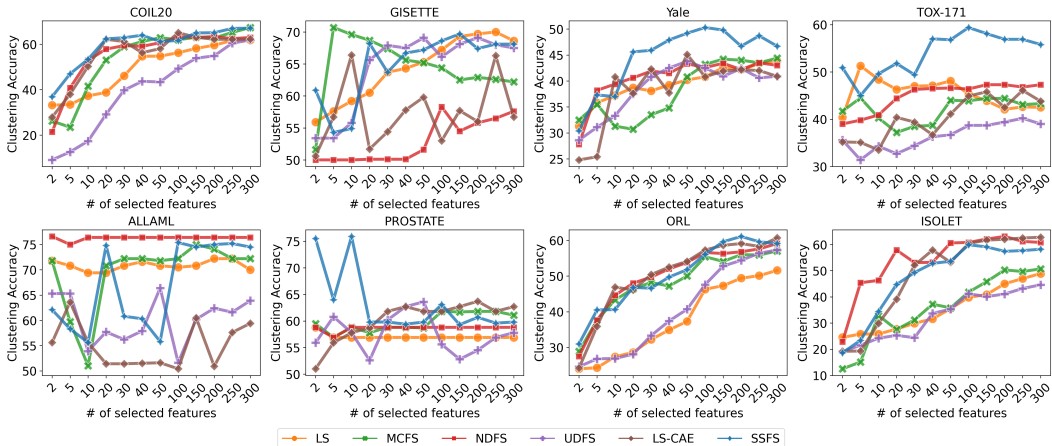

Figure 3: Clustering accuracy vs. the number of selected features on eight real-world datasets.

five features and a visualization of the covariance matrix. We want to identify the features which discriminate between the two blobs.

As Figure 4a demonstrates, the two clusters are linearly separated by three distinct features. Furthermore, examining Figure 4c reveals that while the fourth eigenvector distinctly separates the clusters, the higher-ranked eigenvectors do not exhibit this behavior. This pattern arises due to the correlated noise, significantly influencing the graph structure. The evaluation on this dataset is performed by calculating the true positive rate (TPR) with respect to the top-3 selected features and the discriminative features sampled from the two Gaussian blobs. The performance on the real-world datasets is measured similarly to Section 4.1.

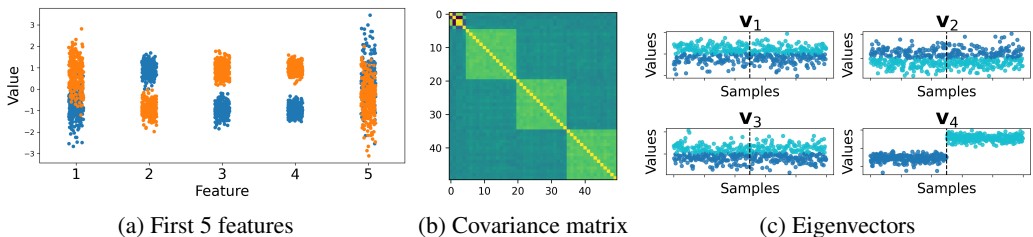

(a) First 5 features      (b) Covariance matrix      (c) Eigenvectors

Figure 4: Visualizations of the synthetic data: Figure 4a: scatter plot of the first five features corresponding to the Gaussian blobs, colored by the real label. Figure 4b: the covariance matrix of the dataset. Figure 4c the top-4 eigenvectors, samples are sorted by the label and are partitioned by the vertical bar, colored according to the output of $k$-medoids.

### 4.2.2 RESULTS

**Eigenvector Selection.** We compare to a variation of SSFS denoted by SSFS (no selection), where we don't filter the eigenvectors and train the surrogate feature selector model on leading $k$ eigenvectors, where $k$ is set to be the number of distinct classes in the data. As can be seen in Figure 5b, our eigenvector selection scheme provides an advantage in seven out of eight datasets. Similarly to Sec. 4.1, filtering the eigenvectors is especially advantageous on the Prostate-GE dataset, as our method successfully selects the most discriminative eigenvectors (see Figure 2a ). On the synthetic dataset, the selection procedure provides a large advantage, as seen in Table 3. As demonstrated in Figure 4c, the fourth eigenvector is the informative one with respect to the Gaussian blobs. Indeed, the fourth eigenvector is selected by the selection procedure, along with the third eigenvector. This eigenvector yields better features compared to MCFS and SSFS (no selection), which rely on the top two eigenvectors.

**Classification and regression.** We compare the following regression variants of SSFS (denoted by SSFS (regression), which use the original continuous eigenvectors as pseudo-labels (without binarization).

- SSFS (regression): uses ridge regression for eigenvector selection and XGBoost regression for the feature selection as surrogate models.

- SSFS (no selection, regression): uses the top $k$ eigenvectors without binarization and XGBoost regression.

As seen in Figure 5a and Table 3, SSFS performs best on six of the eight real-world datasets. Interestingly, when using continuous regression as a surrogate model, the selection procedure does not seem to provide an advantage compared to no selection.

**Complex nonlinear models as surrogate models.** We compare to a variant of our method denoted SSFS (no XGBoost), which employs a Logistic Regression instead of XGBoost as the surrogate feature selector model. Figure 5b shows that XGBoost provides an advantage compared to the linear model on real-world datasets. On the synthetic dataset, the linear variant provides better coverage for the top-3 features that separate the Gaussian blobs, compared to XGBoost (see

Table 3: Synthetic data results: Top-3 selected features (sorted in descending order by rank), along with their TPR (relative to the first five features).

| Method | Top-3 Features | TPR |
|---|---|---|
| SSFS | 2, 9, 19 | 0.3 |
| (no XGBoost) | 4, 3, 2 | 1.0 |
| (no selection) | 43, 30, 49 | 0.0 |
| (regression) | 15, 17, 14 | 0.0 |
| MCFS | 47, 7, 43 | 0.0 |

Table 3 and Figure 4a). That is not surprising since, in this example, the cluster separation is linear in each informative feature. We note, however, that the top-ranked feature by SSFS with XGBoost is a discriminative feature for the clusters in the data (see Figure 4a); therefore, its selection can still be considered successful in the case of a single feature selection.

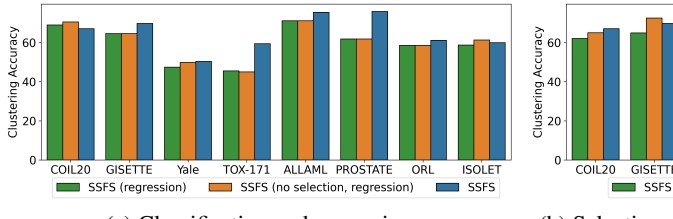 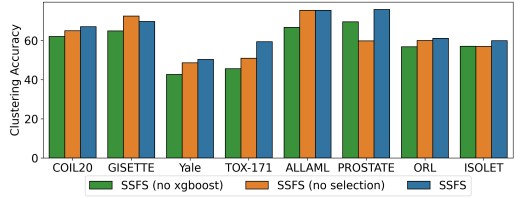

(a) Classification and regression      (b) Selection, and no XGBoost (logistic regression)

Figure 5: Ablation study results on the real-world datasets. The best clustering accuracy over the number of selected features is shown for each method.

## 5 DISCUSSION AND FUTURE WORK

In this work, we proposed a simple procedure for filtering eigenvectors of the graph Laplacian and demonstrated that such a filtration procedure could have a significant impact on the outcome of the feature selection process. Our selection process is based on the stability of a classification model in predicting binary pseudo-labels. However, additional criteria, such as the accuracy of a specific model or the overlap of the chosen features for different eigenvectors, may provide information on the suitability of a specific vector for a feature selection task. Additionally, we illustrated the utility of expressive models, typically used for supervised learning, in unsupervised feature selection.

Another direction for further research is using self-supervised approaches for *group feature selection* (GFS) (Sristi et al., 2022). In contrast to the standard feature selection task where the output is sparse, GFS aims to uncover groups of features with joint effects on the data. Learning models based on different eigenvectors may provide information about group effects with potential applications such as detecting brain networks in Neuroscience and gene pathways in genetics.

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

## A  APPENDIX

### A.1  ABLATION STUDY ADDITIONAL DETAILS

#### A.1.1  SYNTHETIC DATA GENERATION

For the synthetic data, we generated 500 samples, where we used the `make_blobs` function from scikit-learn to generate the first five features, with arguments `cluster_std=1`, `centers=2`.

#### A.1.2  ADDITIONAL EXPERIMENTS INFORMATION

We show here additional information for the ablation study on the real-world datasets, which considers all of the number of selected features range: see Figure 6 and Table 4.

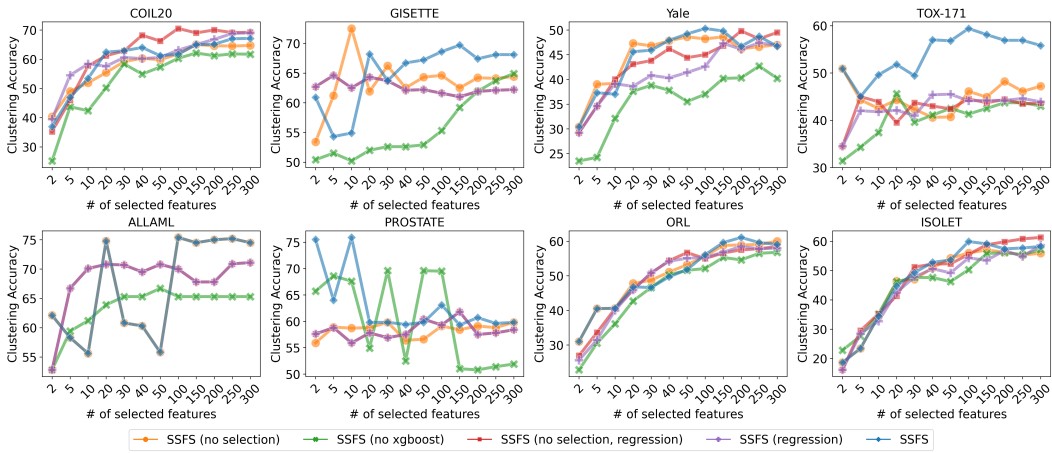

Figure 6: Ablation study: Clustering accuracy on real-world datasets

Table 4: Ablation study: average clustering accuracy on benchmark datasets, the number of selected features is shown in parenthesis for the best clustering accuracy over the feature range.

| Dataset | no selection | no XGBoost | no selection, regression | regression | SSFS |
|---|---|---|---|---|---|
| COIL20 | 65.0 (150) | 62.1 (150) | 70.5 (100) | 69.0 (300) | 67.1 (300) |
| GISETTE | 72.5 (10) | 64.9 (300) | 64.6 (5) | 64.6 (5) | 69.7 (150) |
| Yale | 48.6 (50) | 42.7 (250) | 49.8 (200) | 47.4 (250) | 50.3 (100) |
| TOX-171 | 50.9 (2) | 45.6 (20) | 45.0 (5) | 45.5 (50) | 59.4 (100) |
| ALLAML | 75.4 (100) | 66.7 (50) | 71.1 (300) | 71.1 (300) | 75.4 (100) |
| Prostate-GE | 59.8 (30) | 69.6 (30) | 61.8 (150) | 61.8 (150) | 75.9 (10) |
| ORL | 60.0 (300) | 56.8 (300) | 58.5 (300) | 58.5 (200) | 61.1 (200) |
| ISOLET | 57.0 (150) | 57.1 (300) | 61.3 (300) | 58.7 (300) | 59.9 (100) |
| Mean rank | 2.94 | 4.0 | 3.0 | 3.5 | 1.56 |
| Median rank | 2.5 | 4.5 | 3.5 | 3.5 | 1.25 |

### A.2  IMPLEMENTATION DETAILS

For all datasets, the features are z-score normalized to have zero mean and unit variance.

### A.2.1 HYPERPARAMETERS

For SSFS, on the real-world datasets, we use the same hyperparameters, as follows:

- Number of eigenvectors to select $k$ is set to the distinct number of classes in the specific dataset, they are selected from a total of $d = 2k$ eigenvectors.
- Size of each subsample is 95% of the original dataset.
- 500 resamples are performed in every dataset.
- For the affinity matrix, we used a Gaussian kernel with an adaptive scale $\sigma_i \sigma_j$ such that $\sigma_i$ is the distance to the $k = 2$ neighbor of $x_i$.

In the ablation study, for regression, we use scikit-learn ridge regression (for eigenvector selection) and DMLC XGBoost regressor (for the final feature scoring) with their default hyperparameters.

For all of the baseline methods, we used the default hyperparameters. So, for all methods, including SSFS, the hyperparameters are fixed for all datasets (excluding parameters that correspond to the number of features to select and the number of clusters).

For LS, MCFS, UDFS and NDFS we used an implementation from the scikit-feature library [2] and inputted the same similarity matrices as SSFS for the methods which accepted such an argument. We fixed a bug in MCFS implementation to choose by the max of the absolute value of the coefficients instead of the max of the coefficients (this improved MCFS performance). For LS-CAE, we used an implementation from [3].

---

[2]https://github.com/jundongl/scikit-feature
[3]https://github.com/jsvir/lscae

