# OpenReview forum: "Spectral Self-supervised Feature Selection"
_ICLR.cc/2024/Conference — ICLR 2024 Conference Withdrawn Submission_

### Official Review · Reviewer_Lxsn · 2023-10-30

**Soundness:** 3 good
**Presentation:** 3 good
**Contribution:** 2 fair
**Rating:** 3
**Confidence:** 4

**Summary:**

This paper proposes an unsupervised feature selection method by improving the Laplacian score. It selects k most stable eigenvectors to generate the robust pseudo-labels. Then it uses the pseudo-labels to guide the feature selection.

**Strengths:**

1. The idea of selecting the k most stable eigenvectors is convincing.
2. The paper is well-written and easy to follow.

**Weaknesses:**

1. The paper focuses on a traditional unsupervised feature selection task and tries to improve a classical feature selection method Laplacian score. I'm not sure whether this paper can attract wide interest in learning representations community. Maybe this paper is more approriate to the ICML community.
2. The compared methods are out of date. Since feature selection is a very classical task in machine learning, there are a lot of feature selection methods proposed every year. However, most of the compared methods are proposed about 10 years ago, which cannot demonstrate the effectiveness of the proposed method. The paper should compare with plenty of most recently state-of-the-art methods.
3. It is simple to give an example to show that use the k leading eigenvectors is suboptimal. But could you provide some more theoretical analysis about the selected k eigenvectors of the proposed method?

**Questions:**

See above.

---

### Official Review · Reviewer_v3VG · 2023-10-30

**Soundness:** 3 good
**Presentation:** 2 fair
**Contribution:** 3 good
**Rating:** 6
**Confidence:** 3

**Summary:**

The paper proposed a spectral self-supervised feature selection approach that tests the suitability of each eigenvector obtained from the adjacency/similarity matrix. The eigenvectors are binarized to test the dependence between the eigenvector and the data via feature scoring from surrogate models. Once "relevant" eigenvectors are selected, their binarized versions are used to obtain a feature relevance score for feature selection, again using independent surrogate models. The method is shown to select relevant eigenvectors that may not be among the top choices from the eigendecomposition, which occurs in particular when outliers or structured noise is present in the data.

**Strengths:**

The paper shows the effect in the standard eigenvector-based approach for feature selection of the presence of outliers (using real datasets) and the presence of structured noise (using synthetic data) on the quality of the embedding, highlighting the role of lower-order eigenvectors and the value of binarization in their evaluation. The numerical results (clustering accuracy) and ablation study highlight the positive effect of the proposed additional steps for these cases.

**Weaknesses:**

The presentation is not clear at times. For example, it takes a while to understand the motivation of the approach, which is done a bit by "reverse-engineering": once it is clear that a methodology to evaluate eigenvectors independent of the eigenvalues is established, the motivation for such a choice is presented via experiments. This could be more clearly stated in the introduction by contrasting what is being proposed here (at a high level) with the approaches from the literature.

The proposed approach can be judged as ad-hoc; it is not clear when is it necessary to search eigenvectors beyond the top choices.

The flexibility of the proposed approach leaves some questions that are not fully addressed, such as how to select proxy models to use in each one of the two steps where they are involved.

Minor comments:

In Section 2.1 ISOMPAS should be ISOMAP.
Algorithm 1 line 1: should "pseudo pseudo-labels" be "pseudo-labels"? And "features variance" be "feature variances"?

**Questions:**

If the dataset exhibits standard structure captured by the top eigenvectors, will the method provided here identify the top eigenvectors as the relevant ones as well?

The performance improvement in Figure 3 is not as evident as those provided in Table 2. Can the authors discuss more about the variety of behaviors seen here for SSFS?

---

### Official Review · Reviewer_CaMn · 2023-10-31

**Soundness:** 2 fair
**Presentation:** 3 good
**Contribution:** 3 good
**Rating:** 5
**Confidence:** 3

**Summary:**

The paper introduces a novel feature selection technique based on filtering eigenvectors of the graph Laplacian. Through the utilization of surrogate models, the approach refines feature selection, leading to improved clustering accuracy on multiple real-world datasets. Future research directions point towards harnessing self-supervised methods for group feature selection and exploring their potential applications in fields like neuroscience and genetics.

**Strengths:**

1. **Originality**:
    - The paper presents a fresh take on feature selection by introducing a method grounded in the filtration of eigenvectors of the graph Laplacian. Such an approach stands out due to its uniqueness in leveraging spectral properties for self-supervised feature selection.
    - The utilization of surrogate models as a part of the selection process adds another layer of novelty, as this is seldom seen in traditional feature selection mechanisms.

2. **Quality**:
    - The synthetic data example provides a clear-cut illustration of the technique's ability to discern between higher ranked features influenced by noise and lower-ranked but more useful features, emphasizing the method's practical relevance.

3. **Clarity**:
    - The paper is well-structured with lucid explanations, enabling readers to grasp the core concepts easily. The flow from introduction to experimental results is seamless, making the reading experience smooth.
    - The inclusion of visual aids and intuitive examples, like the aforementioned synthetic data example, greatly augments the paper's clarity.

---

Overall, this paper presents a commendable blend of original ideas and rigorous empirical evaluation. It aptly showcases the potential of spectral methods in the domain of feature selection, thereby enriching the repertoire of techniques available to the research community.

**Weaknesses:**

1. **Absence of Theoretical Justification:** The paper lacks rigorous theoretical analysis supporting its central claim, specifically that "certain features might not significantly affect supervised learning, they can heavily impact unsupervised tasks." In the absence of a theoretical framework, the claim rests primarily on empirical observations. To further the contribution, the authors should delve into a theoretical discussion, offering insights into why and under what conditions their claim holds.

2. **Ambiguity in Real-World Data Analysis:** While the paper demonstrates superior performance against other models by potentially extracting lower-ranked but more informative features, it does not provide direct evidence of this assertion in its real-world-data experiments. A more thorough investigation or a controlled experiment showing how these "lower-ranked but more useful features" contribute to better performance would substantially strengthen the paper's claims.

3. **Inconsistencies in Benchmark Results:** The discrepancies between the accuracies reported in this paper and those in prior works like "Nonnegative Discriminative Feature Selection (NDFS)," specifically for datasets like GISETTE, raise concerns about the reproducibility and fairness of the comparisons. The significant difference in accuracy (66.4% in this paper vs. 74.4% in the NDFS paper) cannot be dismissed as a minor discrepancy. It's crucial to clarify if the datasets used in both papers were identical, if any pre-processing steps differed, or if there were variations in experimental settings. The authors must either provide an explicit justification for these discrepancies or conduct their experiments under the same conditions as prior works to ensure a fair comparison.

**Questions:**

1. **Performance of Lower Ranked Features:** One of the claims in your paper is that the model can extract lower-ranked yet more useful features. Under what specific conditions or scenarios can these lower-ranked features outperform the higher-ranked ones? Is it possible to encapsulate this observation in a more formal manner, perhaps through a specific theorem or lemma that sheds light on this phenomenon?

2. **Discrepancies in Experimental Results:** I noticed that there are some discrepancies in the reported results for certain models, such as NDFS on the GISETTE dataset, when compared to their original publications. Can you provide insights into why such discrepancies arise even when the same model is evaluated on the same dataset? Were there any differences in the experimental setups, data preprocessing steps, or other factors that might have influenced these results?

---

### Official Review · Reviewer_thgm · 2023-11-01

**Soundness:** 3 good
**Presentation:** 2 fair
**Contribution:** 2 fair
**Rating:** 3
**Confidence:** 4

**Summary:**

This paper introduces a graph-based technique for unsupervised feature selection. The method centers on deriving pseudo-labels from the graph Laplacian’s eigenvectors. A subset of eigenvectors is selected based on model stability, and the importance of each feature is estimated by training a surrogate model to forecast the pseudo-labels. Experiments are conducted on several real-world datasets.

**Strengths:**

This paper presents a new method for feature selection.

Experimental results indicate the method's effectiveness.

**Weaknesses:**

The proposed method builds on the Multi-Cluster Feature Selection (MCFS) framework, where feature selection relies on predicting pseudo-labels from spectral clustering. However, its distinct contributions compared to existing MCFS methods remain ambiguous.


On Page 5, the mathematical formulations, including the definition and indexing of $s_m$ are unclear. I am also sure the distinctions between $h_{i, b}$ and $h_i$ and the differences between $s_m$ and $\bar{s}_m$. Additionally, in Section 3.3, I do not understand how the $k$ models are trained how they relate to each other, and their relevance to feature selection.It is also challenging to understand the rationale behind the definition of $score(m)$.

In the experiments, image and speech data are tested. I wonder whether a feature extraction method is applied to the data in the pre-processing. I doubt whether a feature selection method would work on the raw image and speech data.

**Questions:**

1. What are the contributions of the proposed method compared to existing MCFS methods?

2. What is the definition of $s_m$ and what does the index $m$ mean?

3. Waht are the distinctions between $h_{i, b}$ and $h_i$ and the differences between $s_m$ and $\bar{s}_m$?

4. How the $k$ models are trained? How they are related to each other? How are the models used in the feature selection process?

5. What is the rationale behind the definition of $score(m)$.

6. Is a feature extraction method applied in data pre-processing?

---

### Official Review · Reviewer_vbcV · 2023-11-02

**Soundness:** 2 fair
**Presentation:** 3 good
**Contribution:** 2 fair
**Rating:** 3
**Confidence:** 3

**Summary:**

The paper proposes a new method for unsupervised feature selection, which is evaluated with the down-stream task of clustering.
Like similar previous methods, the proposed algorithm gives a score to each feature, based on a synthetic (self-supervised) labeling task.
The novelty is in the selection of the labels, and the model that is used for solving the self-supervised task. In contrast to previous methods, the authors find eigenvectors of the graph laplacian which are robust to resampling, and create pseudo-labels that are robust to outliers in the data.
The evaluation is done using a set of 8 datasets, and for each dataset and method combination, growing subsets of features from size 2 to 300 are extracted, and clusters repeatedly with K-Means, assuming the true number of labels is known, and evaluated using cluster accuracy.
The authors show that their methods performs competitively with other methods for unsupervised feature selection.

**Strengths:**

The problem that is addressed is one of the fundamental problems in machine learning, though one that has not received as much attention as related problems. The authors are motivated by a medical use-case and seem to succeed to improve existing methods to be more robust to outliers in data.
The explanation of the method is easy to follow and the paper is organized well.

**Weaknesses:**

I have several concerns about the evaluation method. The paper claim to follow Cai et. al, but to me there seem to be important differences between their methodology and the reference, though their method seems closer to Li 2012, which I also think uses somewhat questionable evaluation metrics.

Evaluating unsupervised methods is quite difficult, and the paper makes a good effort for a fair and thorough evaluation, but parts of the method are unclear and other parts seem questionably.
In particular, I would prefer the use of adjusted rand index or possibly normalized mutual information instead of clustering accuracy, as more standard metrics.
Second, because of the randomness involved in many aspects of the algorithm, I would appreciate it if there were repetitions of the results, and standard deviations reported. Using the maximum performance over several different numbers of features increases the potential variability of results.
I did a simple experiment to understand the nature of the numbers, on the TOX-171 dataset. Randomly sampling an ordering of features, I used the same evaluation methodology used in the paper, and obtained results between an accuracy of 40% and 48% depending on the run. This means that completely random selection is likely indistinguishable from half of the methods considered on this dataset (though the proposed method is performing much better).
If the algorithms are re-run for each number of features, which is not clear to me from the paper, the baseline would be picking a new random order for each of the number of features. This produces relatively robustly a score of 45% on this dataset, beating two of the competing algorithms.
I'm uncertain whether the protocol of picking the maximum over different numbers of features is a good practice, but given that, the paper should make clear if the algorithm is re-run for each number of features, or if the order of features is fixed.
Also, the paper should ideally include the scores of a random baseline, and an oracle result, i.e. what is the best possible result that can be achieved (say using the real labels as pseudo labels - another more expensive and potentially better scoring oracle would be to go over all subsets and run k-Means but that would be prohibitive). Interestingly, SSFS is close to the oracle performance on TOX-171 based on my analysis, which is quite impressive and somewhat surprising to me.

Finally, the paper does not seem to explain the selection of the kernel hyper-parameters. This is quite critical, in particular the gamma parameter when using the rbf kernel. It's not clear to me what practice I would propose for that, but whatever methods was used should be described in the paper.

minor points:
In the second paragraph of the introduction, "lack of labels" should probably be "absence of labels".
I think section 3.1 could be shortened since the notation is not used afterwards, and in essence you are stating that in a disconnected graph, the eigenvectors to eigenvalue 0 of the laplacian are indicator functions of the connected components.

**Questions:**

Why did you pick cluster accuracy over ARI or NMI for evaluation?
Did you consider spectral clustering with CPQR cluster assignment (Damle et. al. 2019) for creating the pseudo labels?